# Women empowerment through involvement in community-based health and nutrition interventions: Evidence from a qualitative study in India

**Manas Ranjan Pradhan**[1]**, Sayeed Unisa**[1]*****, Ramu Rawat**[2]**, Somila Surabhi**[1]**, Abhishek Saraswat**[2]**, Reshmi R. S.**[1]**, Vani Sethi**[3]

**1** International Institute for Population Sciences, Mumbai, India, **2** Swabhimaan project, International Institute for Population Sciences, Mumbai, India, **3** UNICEF, Delhi, India

\* sayeed.unisa@iipsindia.ac.in

**Data Availability Statement:** Data cannot be shared publicly because of ethical issues. Data are available from the Swabhimaan program at IIPS,

## Abstract

Women's empowerment is fundamental for realizing unalienable human rights and is vital to sustainable development outcomes. In India, the SWABHIMAAN intervention program was an integrated multi-sectoral strategy to improve girls' and women's nutrition before conception, during pregnancy, and after childbirth. This study assesses the role of self-help-group (SHGs) in improving the effectiveness of community health interventions and its impact on their self-empowerment. Qualitative data gathered through in-depth interviews (IDI) with community-based SHG members involved as Nutrition Friend (Poshan Sakhi-PS) in the SWABHIMAAN program in 2018 was used for analysis. Informed consent procedures were followed, and only those who voluntarily consented to the interview were interviewed. Twenty-five IDIs of purposively selected PSs in three states (Bihar, n = 9; Chhattisgarh, n = 8; and Odisha, n = 8) were analyzed thematically, according to Braun & Clarke (2006). NVivo 12 software was used for organizing and coding data. Three central themes that emerged to explain women's empowerment were (1) Barriers & redressal mechanisms adopted by PS, (2) PS as a change-maker, and (3) Changes in the life of PS. The study found that women perceive themselves as more empowered through involvement in the SWABHIMAN intervention program, besides improving the community's and their households' nutritional status. The results suggest that policies and programs on health and nutrition interventions need to involve peer women from the community, leading to more effective outcomes. Empowering women and closing gender gaps in employment/work are critical to achieving the 2030 Sustainable Development Goals.

## Introduction

Women's empowerment is fundamental for realizing unalienable human rights [1] and key to effective and sustainable development outcomes [2]. Investing in women's empowerment can

Mumbai (contact: iipsswabhimaan@iipsindia.ac.in) for researchers who meet the criteria for access to confidential data.

**Funding:** The research received grants under Programme Code & Title- Nutrition 200-Swabhimaan; Project Code & Title: 200-203-03-Swabhimaan, from UNICEF, New Delhi, India. The donor had no role in study design, data collection and analysis decision to publish and preparation of the manuscript.

**Competing interests:** The authors have declared that no competing interests exist.

unlock human potential on a transformational scale [3]. In India, the policy initiatives toward women's empowerment are well in place, especially the involvement of women in development programs, and policy-making at the grassroots is increasingly stressed. The national policy for women's empowerment envisages women attaining their full potential and enabling them to participate as equal partners in all spheres of life and influence social change [4]. The operational strategy includes advocacy and stakeholder partnerships involving and strengthening the grassroots level Self-Help-Group (SHG) capacities, thus, enabling women to participate in policy planning and make informed decisions and choices on social and economic development programs. The Deendayal Antyodaya Yojana-National Rural Livelihoods Mission (DAY-NRLM) also relies on harnessing the innate capabilities of the SHG members and complements them with capacities (information, knowledge, skills, tools, finance, and collectivization) to participate in the growing economy of the country [5]. The theory of change underlying to SHGs includes access to resources (such as increased income, savings, and loan repayments), agency (such as increased autonomy, self-confidence, or self-efficacy), and achievements (such as the ability to transform choices into the desired action) [6–8]. Evidence suggests that gender equality and women's empowerment are vital in stimulating social and economic development [9–11].

Community-based intervention components such as primary education, health education, business or entrepreneurial skills training, awareness of women's rights, or community development training, if added to the SHG program, have a significant positive impact on the SHG members [9]. The effectiveness of community engagement in public health interventions is well-established [12]. Literature shows positive outcomes of peer-led community interventions on health and empowerment [13]. Moreover, group support through training is a critical factor in increasing women's empowerment and, more specifically, improvements in self-confidence [14–18]. Thus, only microfinance or livelihood components may not be sufficient to benefit SHGs in all empowerment domains. Instead, integrating SHGs with other community-based intervention programs, such as health programs, may be necessary. Some of these integrated models, in which health, livelihoods, or other activities are added to savings and credit-focused groups, are increasingly common in South Asia [14, 16]. It is thus imperative to assess the effect of interventions on the health and empowerment of the change-makers implementing community-level interventions, especially in the Indian context. This paper explores the role of SHGs in improving the effectiveness of community health interventions and its impact on their self-empowerment.

The SWABHIMAAN demonstration program was initiated as a collaborative pilot project across five blocks of three Indian states (Bihar, Chhattisgarh, and Odisha), including—women's SHGs federated as Village Organizations (VO) under NRLM to implement village nutrition plans through cash grants. The SWABHIMAAN demonstration program is an integrated multi-sectoral strategy to improve girls' and women's nutrition before conception, during pregnancy, and after childbirth in selected blocks of the states mentioned above. The program entails two strategies (i) formal system strengthening to improve coverage of food security, health, nutrition, water, and sanitation services and (ii) partnering with village organizations to design, implement and monitor a multi-sector program for adolescent girls and women [19]. UNICEF India assists the three States' Livelihood Missions in implementing the integrated strategy. In the SWABHIMAAN program, the system's strengthening was done by coordinating the Departments of Health, Civil Supplies, Social Welfare, and Public Health Engineering, with technical and financial support from UNICEF. A reputed academic institute was the technical partner for the baseline and served as the principal agency for the impact evaluation in the midline and end-line of the program.

The SHG program under NRLM was considered untapped to reach out to adolescents and pregnant women with a unique package of reproductive, health, and nutrition messages and services. Evidence suggests that these community organizations and their federations have the potential to manage grants for improving last-mile delivery of essential nutrition services for women, provided they are enabled, supervised, and provided protection against violence and exploitation [20]. In the context of Indian public health services, the field workers are critical to improving the last-mile delivery of health services and undertaking a range of activities like community mobilization, counseling, and record-keeping, to name a few, which have primarily been considered honorary workers. In the SWABHIMAAN Program, a "Poshan Sakhi" (Nutrition friend/peer) was designated from the SHG members, trained to reach out to adolescents and pregnant women in their working area to deliver the special package of reproductive health and nutrition messages as well as services. This paper intends to qualitatively assess the role of the SWABHIMAAN intervention program (nutrition and health training and implementation) in empowering the SHG members in rural areas of three states of India. The impact evaluation was registered with the Registry for International Development Impact Evaluations [RIDIE-STUDY-ID-58261b2f46876] and the national clinical trials registry [CTRI/2016/11/007482]. The institute IRB approved this study.

## Methods

### Data

This study is based on the qualitative data gathered through in-depth interviews (IDI) with community-based SHG members involved as Poshan Sakhi (PS) in the SWABHIMAAN program. The data was collected during midline evaluation in 2018. The participants were selected purposively from the pool of PSs involved in the intervention. We approached 25 PSs and could interview all of them at their convenient time and place. There was no refusal as they were active partners in the implementation program. Specifically, the present study is based on 25 IDIs of PSs from three Indian States-Bihar (n = 9), Odisha (n = 8), and Chhattisgarh (n = 8). The number of participants is sufficient to draw relevant inferences as it is well above the minimum suggested number to justify data saturation [21–23]. Informed consent procedures were followed, and only those who voluntarily consented to the interview were interviewed. Written consent was sought and obtained from all the participants. The interviews were audio-recorded after written consent from the respondents. Trained research interviewers performed the IDIs using a pre-tested interview guideline. All the interviews were performed in the local languages (Hindi in Bihar and Chhatisgarh, and Odiya in Odisha). The duration of IDIs ranged between 45–65 minutes. The interviewers were from the states under the study but were not from the same village as the participants to avoid bias.

### Analysis

The field notes supplemented the audio-recorded IDIs to finalize the transcriptions. NVivo 12 was used for organizing and coding data. The data were analyzed thematically, according to Braun & Clarke [24]. A theme captures something important about the data concerning the research question and represents some level of *patterned* response or meaning within the data set. Thematic analysis is a method for identifying, analyzing, and reporting patterns (themes) within data. As a method, it further interprets various aspects of the research topic [25]. We familiarized ourselves with data, generated initial codes, searched for themes, reviewed themes, defined and named themes, and produced the report. The coding process started with identifying and coding three IDIs, independently by the first and the corresponding authors. Both authors discussed the initial emerging codes and finalized the code list to ensure inter-coder

reliability. Subsequently, the first author coded the data and briefed the study team about coding categories and emerging themes from time to time. The codes, categories, and themes were finalized in consultation with the study team. The relationship among emerging themes, categories, and codes was studied using memos and mind maps through an iterative process. In this study, anonymous direct quotes were used to substantiate the findings.

### Profile of study participants

The average age of the 25 PSs interviewed was 29 years. Of the 25 PSs, nine had 12+ years of schooling, while the remaining had 8–10 years of schooling. All the PSs were associated with SHGs and worked for the SWABHIMAAN intervention study for two-plus years.

### Results

Three central themes that emerged to explain PS's empowerment through involvement in the intervention study were (1) Barriers & redressal mechanisms adopted by PS, with categories of family, community, and organization level barriers plus effective ways adopted to address those barriers, (2) PS as a change-maker, with categories of malnutrition and sensitizing the community on nutrition and nutritious foods, and improved reproductive health, and (3) Changes in the life of PS, with categories of nutritional knowledge and improved health, recognition in the community, and sense of independence [Fig 1].

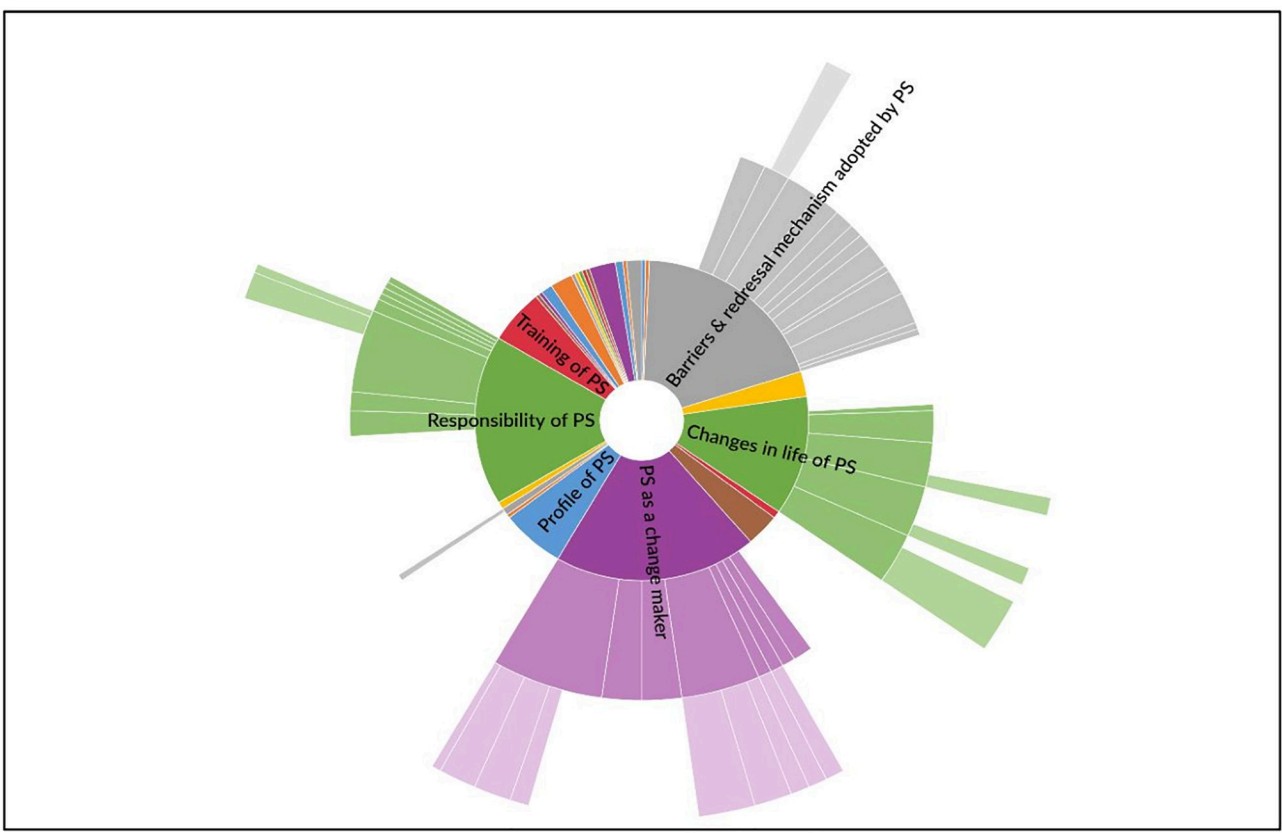

**Fig 1. Key themes emerged from the in-depth interview of the Poshan Sakhis.**

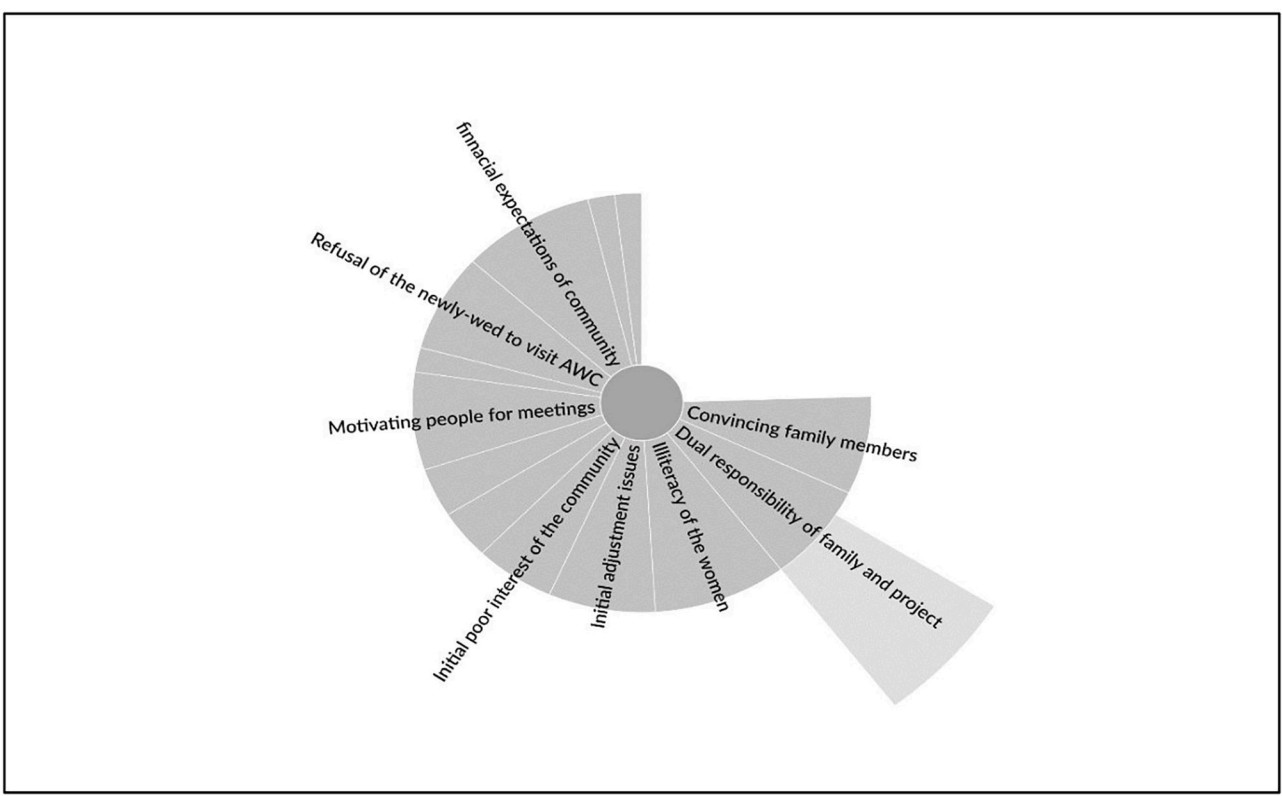

**Fig 2. Barriers & redressal mechanism adopted by Poshan Sakhis.**

## Barriers & redressal mechanism adopted by PS

The PSs encountered various individual, family, community, and organizational level barriers while executing their responsibility [Fig 2]. The first hindrance was to convince the spouse and, for some, both the spouse and the in-laws to get involved in the intervention program. Many got the nod to engage as PS with specific riders. Mainly, they were instructed to only visit within the village and even certain areas/communities. It was also made clear that household chores must be the priority and should not be compromised at any cost. There was even an instance of a PS leaving the program due to concerns of the family.

*She (earlier PS) had to leave the program 4–5 months after getting the training because she had personal and family problems*

*(IDI 9, PS from Bihar).*

Performing the dual duty as a wife/mother and PS was difficult for many PSs, reported as initial interruptions. Some felt overburdened to work as PS and perform the expected household duties.

*I had got problems at our home. I have to prepare food for the family and to attend the meeting. Moreover, women come to the meeting if I call 2–3 times. Time-to-time, I have to tell them. My family members ask who will do household work for you (me) daily*

*(IDI 9, PS from Chhatisgarh).*

*We were not getting adequate time for our work. We were overburdened by lots of household work*

*(IDI 22, PS from Odisha).*

*I have to complete household chores first and then come to work. I have to call other women (didi) for the meeting and bring material from Jagdalpur*

*(IDI 10, PS from Chhatisgarh).*

Some families were further skeptical about the meager financial incentives for the PS, at the cost of perceived negligence of family responsibilities, visiting outside the village, and participating in meetings in various places.

*You (the PS) may have to visit outside the village, how much money will you get?, Who will manage the household work and children*

*(IDI 18, PS from Odisha).*

PSs could overcome the initial concerns on the personal and family front to fulfill their interest and commitment to work as a PS. Most of the PS believed that they gradually got family support due to the in time completion of household duties and their contribution to family income besides recognition in the community. The recognition and role within the community had further cemented their decision-making ability within their household.

*Earlier, I had a problem because of my young child; now, I get the support of my family and the VO for our work. Earlier, we individually worked, but now ASHA, ANM, and AWW work together.*

*(IDI 13, PS from Chhatisgarh).*

*Now, my family is supportive of my work as PS. They feel I am doing good for the village*

*(IDI 20, PS from Odisha).*

*Before, nobody asked my opinion, but now I am being consulted on healthcare decisions in the family. My husband feels I know many things about health matters*

*(IDI 7, PS from Bihar).*

At the community level, the initial poor interest of the community, no support from Panchayat members, inter-caste equation, inability & refusal of the newlywed to visit the Anganwadi center, and financial expectations of community for participation in meetings/awareness programs were the significant hindrances encountered by the PS during the implementation of the SWABHIMAAN program.

*The family does not allow the newlyweds (nayi dulhan) to go out of home or to other parts of the village (next paraa) for meetings*

*(IDI 11, PS from Chhatisgarh).*

*Distance to the Anganwadi center prevents many from attending the VHND. Moreover, as I am from a particular caste, women from other castes avoid meetings organized by me*

*(IDI 14, PS from Chhatisgarh).*

*When we go for a home visit, some women don't talk with us properly, and we feel bad. We explain to them a lot, but they don't understand. They don't cooperate because they don't get any financial help*

*(IDI 1, PS from Bihar).*

The women's illiteracy was further perceived to make it difficult for the PS to convince them to participate in program activities to improve their health and nutrition.

*Our area is tribal, and women are not educated. They need to be oriented several times*

*(IDI 20, PS from Odisha).*

The PSs overcame these community-level barriers through frequent visits to targeted households/women, convincing people about the importance of nutrition, and rescheduling the meeting timing and venue convenient to the community members.

*As newlyweds are not allowed to come out, we go to their homes and talk to them. After convincing her in-laws, we ensure that she participates in the meeting in her paraa (part of her village). We welcome them with coconut and flowers in our meeting*

*(IDI 11, PS from Chhatisgarh).*

*There was little difficulty in doing the meetings. Women started asking for money for it. Then we made them understand that this is for you. We are giving information about your health and good food*

*(IDI 1, PS from Bihar).*

*Now, all understand the importance of the program; they cooperate well. Now ANM and ASHA are also with us, so all women understand. All come for the meetings*

*(IDI 12, PS from Chhatisgarh).*

An organizational challenge encountered by a few PSs was the absence of senior employees in all the meetings/activities organized by the PS. It was viewed that women perceive the meeting attended by higher officials from outside to be of importance and were more likely to join it. Non-cooperation of ASHA, ANM, and AWW in the initial stages of program implementation was also a concern for many across the states. A few also raised shortages and delays in money for conducting activities under the program.

*The main problem is that sometimes we do not get the proper ingredients for certain foods from the market. Sometimes, we also have a shortage of funds and run without money*

*(IDI 9, PS from Bihar).*

However, commitment to responsibility, understanding the importance of the program for own and community health, financial incentives, and recognition in the community led the PSs to continue in the program.

*I feel the program is good for people. I also get recognition in the village*

*(IDI 9, PS from Bihar).*

*When I joined Swabhimaan, I did not know that I was underweight, I attended training where I got to know about nutritious food, and now I'm normal. Because of it now, I want more and more people to understand this*

*(IDI 24, PS from Odisha).*

### PS as a change maker

PSs believed to have successfully brought a positive change in the nutrition and health of the women in the community [Fig 3]. Nutrition hygiene imparted by the PSs, the reduced gender gap in the intra-household consumption of nutritious foods, and nutrition/kitchen gardens initiated through the PS were perceived to have reduced the malnutrition situation in the community.

*Almost all women are now adopting healthy behavior, we told them. Those who are malnourished also adopted it. Now slowly, they are coming out of malnourishment. Changes have happened*

*(IDI 2, PS from Bihar).*

*Before, women don't come to Aarogya Diwas, but now they are coming. There is also a change in farming. They are also doing farming (kitchen garden) in less space. The program is going very well*

*(IDI 22, PS from Odisha).*

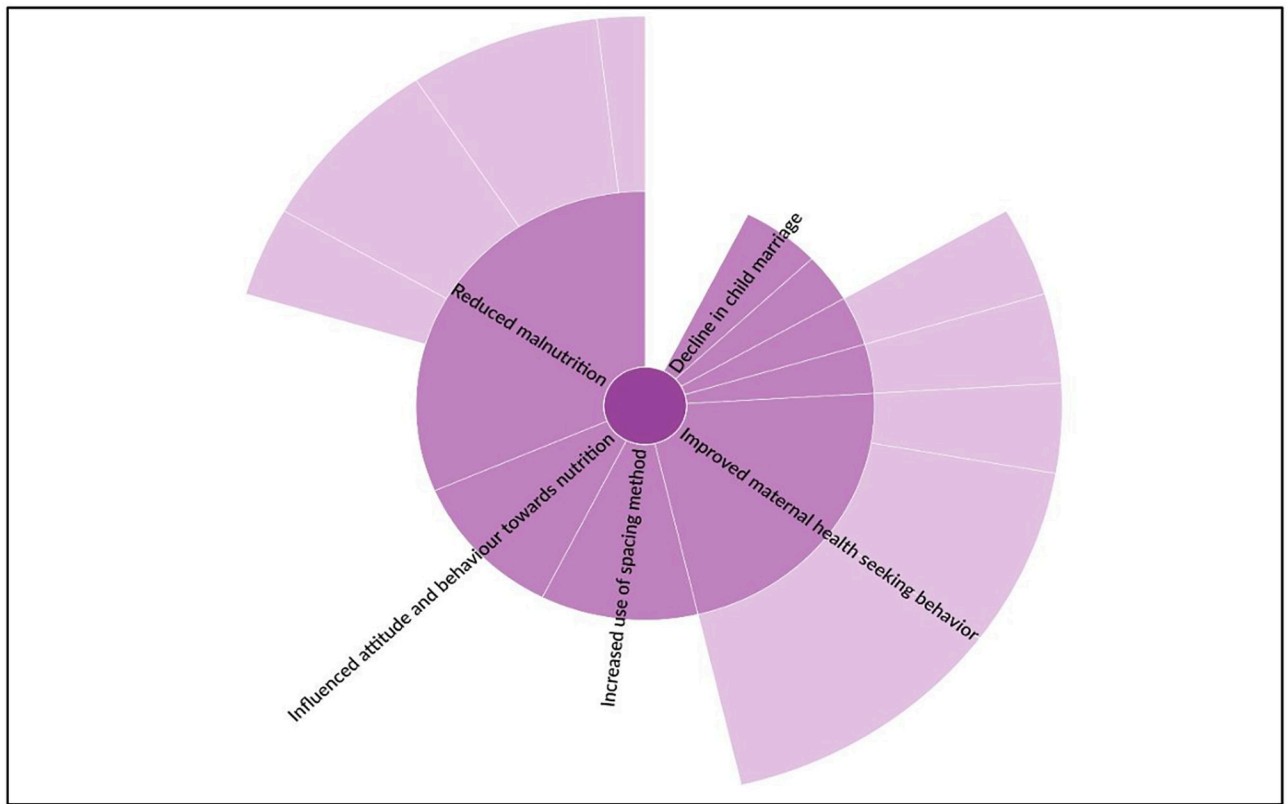

**Fig 3. Poshan Sakhis as a change-maker in the community.**

*A total of three women are now out of undernourishment in my catchment area, and within last six months, one woman came out of malnutrition*

(IDI 3, PS from Bihar).

PSs again perceived to have significantly improved the reproductive health status of women in the community. Most PSs claimed credit for the perceived decline in child marriage and the use of modern spacing methods of contraception among newlyweds. Additionally, almost all the PSs believed they were instrumental in increased pregnancy registration, enhanced antenatal care (ANC), consumption of nutrition supplements, institutional delivery, and on-time child immunization in their areas.

*Now pregnant women are going for ANC checking. They understand what is good for them. They are starting to believe in Poshan Sakhi*

(IDI 22, PS from Odisha).

*Now pregnant women do early registration, receive TT. Earlier, they were shy, now by the third month, they do registration. They are going for hot cooked meal earlier not. Earlier, all were not consuming iron tablets. Now they consume. Now they care for food, water, and hospital delivery*

(IDI 10, PS from Chhatisgarh).

*Earlier, not many women used modern methods of family planning, but now they do. Now they are getting wise*

(IDI 3, PS from Bihar)

## Changes in the life of PS

The involvement of SHG members, such as PSs, in the intervention program brought considerable changes in their lives [Fig 4]. It has enhanced nutritional knowledge and has improved the health-seeking behavior of the PSs.

*There are many changes. Previously I did not know that children should be given supplementary foods, but now I know what a mother should eat and for how long. Now we understand the importance of vaccination, food, etc*

(IDI 5, PS from Bihar).

*I feel happy as I received training on it and got informed that my daughter is an adolescent (Kishori). Now I care for her health and give her food 3–4 times, and with varieties, I would give her one vegetable earlier. Now I prepare three-colour food. I learned to design a kitchen garden. One type of Doctor we are now. I feel happy to be in this role. I see a change in people, so I feel happy. People listen to us and change so that we see and feel happy. Someone's life is changed, so we feel good*

(IDI 11, PS from Chhatisgarh).

PSs perceived to have earned recognition and respect in the community through their work for the intervention study. This resulted in increased self-confidence of the PSs.

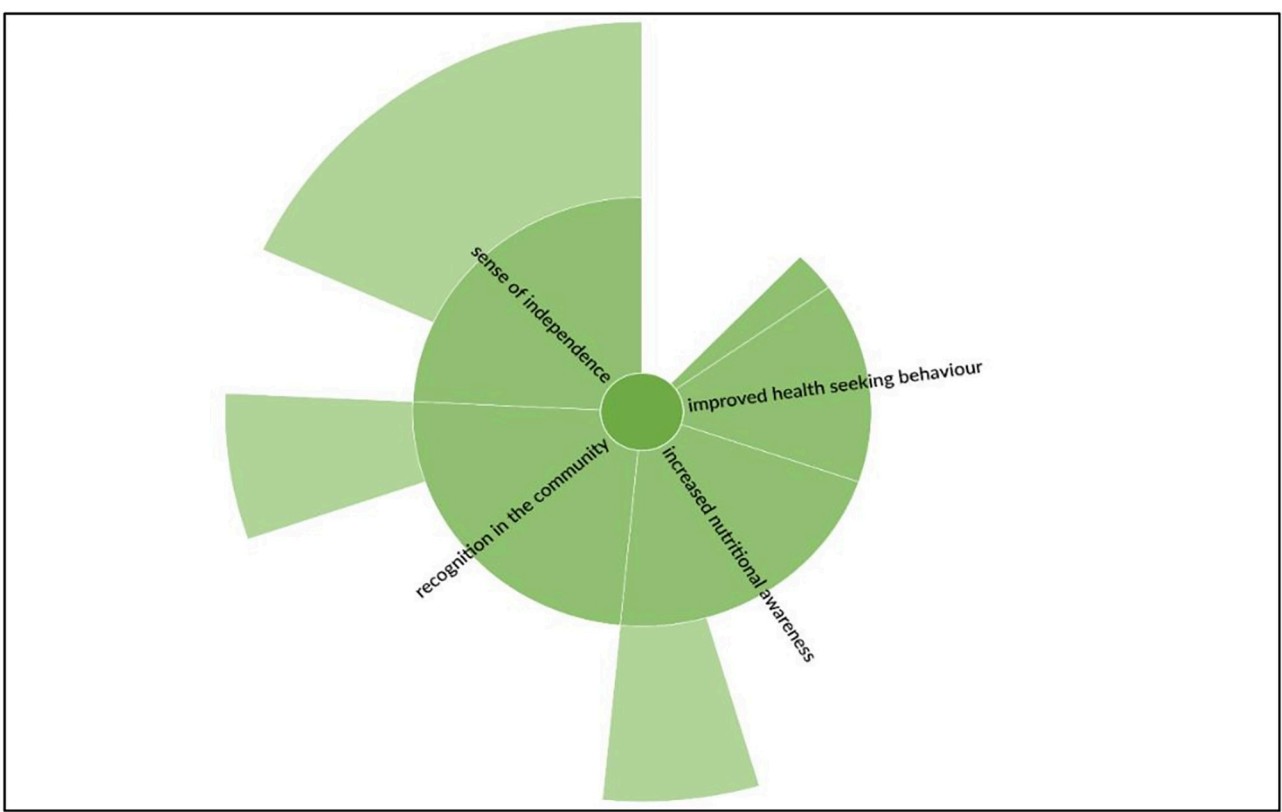

**Fig 4. Changes in the life of Poshan Sakhis.**

*I am happier in my life than past. It makes me happy that people know my name and understand my work. We have to work hard, but when we get respect, it feels good*

(IDI 1, PS from Bihar).

*Before, I was utterly unknown about everything, but now I know everything. Before SWABHI-MAAN, nobody knew us, but now everyone knows us and gives us respect*

(IDI 4, PS from Bihar).

*When I go somewhere, all women try to talk. It feels good to get respect from people. If anything happens, women first inform me*

(IDI 3, PS from Bihar).

*Earlier, I had hesitation to talk, but not now. Now I talk freely. Now even with training sir, I talk freely*

(IDI 9, PS from Chhatisgarh).

*Happy to say that despite the water shortage, people now have a small kitchen garden in their courtyard. Most of them are growing vegetables for their consumption. Two-to-three produce on a large scale and has promoted it like a business*

(IDI 21, PS from Odisha).

It also emerged that the PSs developed a sense of independence and satisfaction. It was due to their involvement and successful execution of the responsibilities resulting in a positive change in the nutrition and health status of the community.

> I was not aware of many issues before Swabhimaan. Earlier, I used to fear what to say and how to talk. Now we can talk about our thoughts and problems to others without hesitation. I feel happy to say that the villagers recognize me as I work for them. Even I give training to another didi (PS) in the next village. I feel happy to train others. I feel now that I am standing on my own feet

> (IDI 9, PS from Chhatisgarh).

### Empowerment of the PS

Involvement in the program has resulted in the overall empowerment of the PSs [Fig 5]. Specifically, it had worked through educational, economic, health, and extensive exposure of the PSs. These women gained knowledge and experience through the program, which helped improve the nutritional and healthcare autonomy of the PSs, resulting in improved health and nutrition for the community and their own family. They were perceived to have strengthened

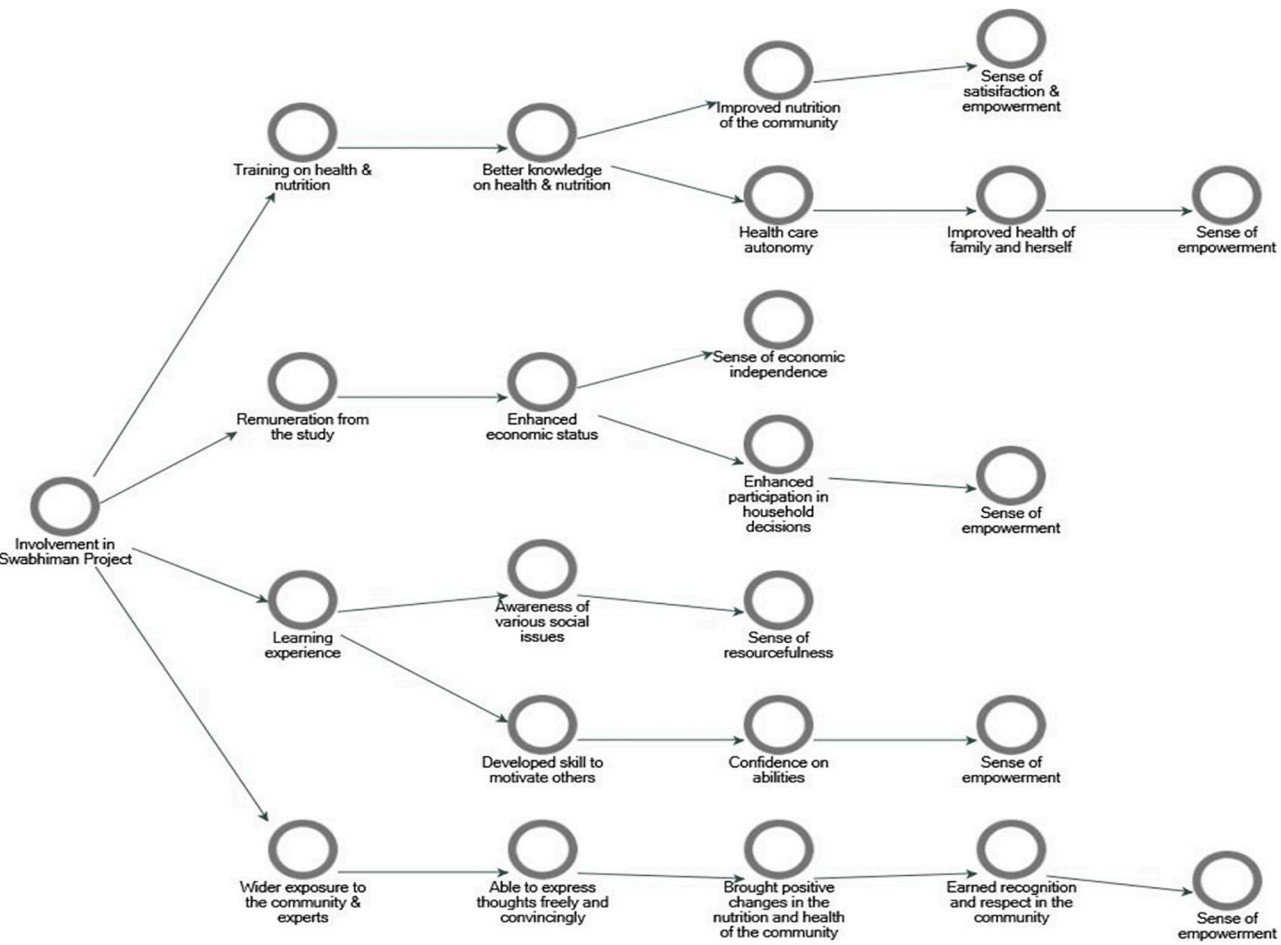

**Fig 5. Mind map of women empowerment through involvement in the study.**

their self-confidence and ability to convince people of critical social issues through exposure to the community and frequent interaction with experts. The respect and recognition resulting from hard work, financial incentives, and enhanced understanding of health, nutrition, and social issues also contributed to a sense of empowerment among the PSs.

## Discussion

The Swabhimaan program involved the SHG women from the community as Poshan Sakhi to implement the intervention. It helped them develop as change agents who positively contributed to young girls' and women's health and nutrition. Enhanced knowledge of health and nutrition care, mobility and opportunity to express views, strengthened skills to communicate and motivate people, recognition in the household and community, and increased economic status have brought a clear sense of empowerment among the PSs. Earlier studies do recognize the promotion of health knowledge [26], physical mobility [27], enhancement of skills for information technology and communication [28], work participation [29], and economic independence [30] as enablers of women's empowerment. It is also proved that employment and income sources empower women through increased domestic decision-making power [31].

The study found that the involvement of PSs as a mediator of knowledge transfer is beneficial, and particularly, the familiarity of the PSs with the community has helped achieve the desired nutritional and health outcomes. The finding confirms an earlier study that found that the peer-led intervention model improves marginalized women's combined health and economic outcomes in India [13]. The PSs could successfully initiate kitchen gardens, which have proven beneficial for addressing malnutrition. A past study reveals the positive impacts of home gardens towards addressing food insecurity and malnutrition, besides benefits such as income and livelihood opportunities for resource-poor families [32]. Nutrient-sensitive agriculture has also been the best approach to improving children's nutritional status [33].

Women's empowerment has been proven to enhance dietary diversity [34–36] and micronutrient deficiency among women [37], besides improving child nutrition [38]. Empowered women are better positioned to enhance food security for children by increasing household food quantity, diversity, and nutritional value. In addition to contributing to the decline in malnutrition among young girls and women in the community, PSs adopted healthy behaviors for themselves and their children, thus reducing malnutrition at home.

The study found a sense of empowerment and decision-making autonomy among PSs after their involvement in the intervention. An earlier study also reveals that empowerment is a significant determinant of healthcare decision-making and women's physical and mental health [39]. Our study indicates higher healthcare autonomy among women, which influenced their health-seeking behavior. The result aligns with an earlier study that found a strong association between healthcare autonomy and maternal healthcare in developing countries [40]. PSs have improved communication skills, networking, self-confidence, and enhanced socioeconomic status. A past qualitative study in Uganda also reveals that participatory community intervention enhances the health and socioeconomic status of the implementing women [41]. The study highlighted the contributory role of involvement in community-level interventions in women's empowerment, a finding with a significant policy and program implication. However, the results are based on qualitative data gathered through IDIs with a purposively selected group of women. Therefore, caution needs to be taken regarding the results' transferability.

## Conclusion

The PSs perceive themselves as more empowered through involvement in the Swabhimaan intervention program, besides improving the nutritional status of the community and their households. The results suggest that policies and programs on health and nutrition interventions need to involve peer women from the community, leading to more effective outcomes. Empowering women and closing gender gaps in employment/work are critical to achieving the 2030 Sustainable Development Goals 1, 2, 3, 5, and 10.

## Acknowledgments

We would like to thank all the community participants who worked in this intervention study and our respondents.

## Author Contributions

**Conceptualization:** Manas Ranjan Pradhan, Sayeed Unisa.

**Data curation:** Manas Ranjan Pradhan, Ramu Rawat, Somila Surabhi, Abhishek Saraswat, Reshmi R. S.

**Formal analysis:** Manas Ranjan Pradhan.

**Funding acquisition:** Sayeed Unisa, Vani Sethi.

**Methodology:** Sayeed Unisa, Ramu Rawat, Somila Surabhi, Abhishek Saraswat, Reshmi R. S.

**Project administration:** Sayeed Unisa, Vani Sethi.

**Software:** Manas Ranjan Pradhan.

**Supervision:** Sayeed Unisa.

**Writing – original draft:** Manas Ranjan Pradhan.

**Writing – review & editing:** Sayeed Unisa, Ramu Rawat, Somila Surabhi, Abhishek Saraswat, Reshmi R. S., Vani Sethi.

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
