## [Decision Letter · Decision Letter 0]

14 Feb 2023

PONE-D-22-25775Women empowerment through involvement in community-based health and nutrition interventions: Evidences from a qualitative study in IndiaPLOS ONE

Dear Dr. Unisa,

Thank you for submitting your manuscript to PLOS ONE. After careful consideration, we feel that it has merit but does not fully meet PLOS ONE’s publication criteria as it currently stands. Therefore, we invite you to submit a revised version of the manuscript that addresses the points raised during the review process.

Thank you for submitting this important manuscript. Please find attached a peer review comments. Please work on this, revise and resubmit. 

We look forward to receiving your revised manuscript.

Kind regards,

Vijayaprasad Gopichandran

Academic Editor

PLOS ONE

Reviewers' comments:

Reviewer's Responses to Questions

**Comments to the Author**

1. Is the manuscript technically sound, and do the data support the conclusions?

Reviewer #1: Yes

2. Has the statistical analysis been performed appropriately and rigorously? 

Reviewer #1: N/A

3. Have the authors made all data underlying the findings in their manuscript fully available?

Reviewer #1: No

4. Is the manuscript presented in an intelligible fashion and written in standard English?

Reviewer #1: Yes

5. Review Comments to the Author

Reviewer #1: General comments: This is an important topic reflecting on community participation of women from SHGs and its impact on their empowerment. The three themes derived from the data are presented well however, as this paper primarily focuses on empowerment of SHGs, the barriers at personal and professional front could have been probed more in terms of:

1. Challenges at the level of family: Family member’s positive or negative attitude towards their job (especially of the spouse and in-laws ), how it influences their work? and how do they overcome such challenges? Does the recognition and role within community, has empowered their decision making ability within their household? (As decision making ability within household is an important indicator of women empowerment). Insight on such aspects will shed more light on the crucial role of family in women empowerment.

2. Challenges at the organization level: - Factors like hierarchical support/conflicts/discrimination, issues related to delay in incentive dispersal (This is a common issue faced by HCWs) and what motivates them to continue their work despite such challenges, if any.

Some minor comments:

The data collection method has been presented well except for few points that needs modification or further information needs to be added to make it more streamlined with qualitative study guidelines:

1. Line 154-156: Coding of data requires two or more independent researchers ensuring Intercoder Reliability (ICR), kindly provide the information on number of researchers involved in coding apart from the primary author.

2. The number of respondents who were approached, how many refused to participate and reason for refusal.

3. Was the transcript shared with the participants for comments or correction?

4. Duration of IDIs

Line 94-95: “This paper attempts to examine the impact on the implementors and their role in implementing the intervention in enhancing empowerment”:

This line could be modified as it confuses whose empowerment authors are referring to, SHGs or community? Since, the participants are only PSs (Poshan Sakhis) it would be better to put the rationale in their terms.

Suggestion: “ This paper explores the role of SHGs in improving the effectiveness of community health interventions and its impact on their self empowerment.

Line 248: “registration, enhanced antenatal care (ANC), consumption of IFA tablets/syrup”

Suggestion: Kindly mention consumption of nutrition supplements or mention the full range of supplements given i.e IFA, Calcium and Albendazole tablets.

Overall the paper provides a good insight on the role of PSs in community health intervention and how it is beneficial for their personal growth. If the authors can respond to the above queries, then this paper may be considered for publication.

6. PLOS authors have the option to publish the peer review history of their article (what does this mean?). If published, this will include your full peer review and any attached files.

Reviewer #1: **Yes: **Isha Sinha

---

## [Author Response · Author response to Decision Letter 0]

24 Mar 2023

Response to review and editor comments

Comments Response

R- The revised manuscript follows the suggested style requirements

R-The method section includes additional details regarding type of participant consent.

R-The research received grants under Programme Code & Title- Nutrition 200-Swabhimaan; Project Code & Title: 200-203-03-Swabhimaan, from UNICEF, New Delhi, India. The aforementioned grant number is provided in the Funding information Section in the cover letter and online financial disclosure section.

R- The revised manuscript contains the following data availability statements.

“Data cannot be shared publicly because of ethical issues. Data are available from the Swabhimaan program at IIPS, Mumbai (contact: iipsswabhimaan@iipsindia.ac.in) for researchers who meet the criteria for access to confidential data”.

R-The reference list has been checked and found to be complete and correct.

Reviewer 1 

Reviewer #1: General comments: This is an important topic reflecting on community participation of women from SHGs and its impact on their empowerment. 

The three themes derived from the data are presented well however, as this paper primarily focuses on empowerment of SHGs, the barriers at personal and professional front could have been probed more in terms of:

1. Challenges at the level of family: Family member’s positive or negative attitude towards their job (especially of the spouse and in-laws ), how it influences their work? and how do they overcome such challenges? Does the recognition and role within community, has empowered their decision making ability within their household? (As decision making ability within household is an important indicator of women empowerment). Insight on such aspects will shed more light on the crucial role of family in women empowerment.

2. Challenges at the organization level: - Factors like hierarchical support/conflicts/discrimination, issues related to delay in incentive dispersal (This is a common issue faced by HCWs) and what motivates them to continue their work despite such challenges, if any. 

R-Thank you for reviewing the paper and the valuable comments to enhance its quality. As appreciated, efforts were made to present the three emerging themes from the data. Nevertheless, in view of the suggestions, we have made revised the manuscript highlighting the barriers at personal and professional front.

Some minor comments: 

The data collection method has been presented well except for few points that needs modification or further information needs to be added to make it more streamlined with qualitative study guidelines: 

R-As suggested, we have done required changes to make data collection method more streamlined

1. Line 154-156: Coding of data requires two or more independent researchers ensuring Intercoder Reliability (ICR), kindly provide the information on number of researchers involved in coding apart from the primary author. 

R-Both the first and the corresponding author were involved in initial coding of 3 IDIs. The codes were discussed at length and a comprehensive code list was prepared in consensus, to ensure ICR. Specifically, the following sentences are added in the revised manuscript.

“The coding process started with identification and coding of three IDIs, independently by the first and the corresponding author. To ensure, inter-coder reliability, both the authors discussed the initial emerging codes and finalized the code list. Subsequently, the first author coded the data and briefed the study team about coding categories and the emerging themes from time to time.”

2. The number of respondents who were approached, how many refused to participate and reason for refusal. 

R-We could interview all the chosen respondents. Specifically, the following sentences are added in the revised manuscript. “We approached 25 PSs and could interview all of them at their convenient time and place. There was no refusal as they were active partners in the implementation program. Specifically, the present study is based on 25 IDIs of PSs from three Indian States-Bihar (n=9), Odisha (n=8), and Chhattisgarh (n=8)”.

3. Was the transcript shared with the participants for comments or correction? 

R- Individual respondents were briefed on the points raised/told by them to avoid any misunderstanding/assessment by the interviewer. However, the transcripts which were prepared after the interviews were not shared with them. 

4. Duration of IDIs 

R- The duration of IDIs ranges between 45- 65 minutes. 

Line 94-95: “This paper attempts to examine the impact on the implementors and their role in implementing the intervention in enhancing empowerment”:

This line could be modified as it confuses whose empowerment authors are referring to, SHGs or community? Since, the participants are only PSs (Poshan Sakhis) it would be better to put the rationale in their terms.

Suggestion: “ This paper explores the role of SHGs in improving the effectiveness of community health interventions and its impact on their self-empowerment. 

R- Appreciate the suggestion. We have modified the sentence as suggested.

Line 248: “registration, enhanced antenatal care (ANC), consumption of IFA tablets/syrup”

Suggestion: Kindly mention consumption of nutrition supplements or mention the full range of supplements given i.e IFA, Calcium and Albendazole tablets. 

R- As suggested, we have modified the sentence

Overall the paper provides a good insight on the role of PSs in community health intervention and how it is beneficial for their personal growth. If the authors can respond to the above queries, then this paper may be considered for publication. 

R- Thank you for appreciating the contribution of this paper. We have incorporated all the comments of the reviewer in the revised manuscript to be considered for publication. 

Thank you

---

## [Editor Report · Decision Letter 1]

3 Apr 2023

Women empowerment through involvement in community-based health and nutrition interventions: Evidence from a qualitative study in India

PONE-D-22-25775R1

Dear Dr. Unisa,

We’re pleased to inform you that your manuscript has been judged scientifically suitable for publication and will be formally accepted for publication once it meets all outstanding technical requirements.

Kind regards,

Vijayaprasad Gopichandran

Academic Editor

PLOS ONE
---

## [Editor Report · Acceptance letter]

11 Apr 2023

PONE-D-22-25775R1 

Women empowerment through involvement in community-based health and nutrition interventions: Evidence from a qualitative study in India 

Dear Dr. Unisa:

I'm pleased to inform you that your manuscript has been deemed suitable for publication in PLOS ONE. Congratulations! Your manuscript is now with our production department. 

Kind regards, 

on behalf of

Dr. Vijayaprasad Gopichandran 

Academic Editor

PLOS ONE